# DAVA: Disentangling Adversarial Variational Autoencoder

**Benjamin Estermann**
ETH Zürich
Switzerland
estermann@ethz.ch

**Roger Wattenhofer**
ETH Zürich
Switzerland
wattenhofer@ethz.ch

## Abstract

The use of well-disentangled representations offers many advantages for downstream tasks, e.g. an increased sample efficiency, or better interpretability. However, the quality of disentangled interpretations is often highly dependent on the choice of dataset-specific hyperparameters, in particular the regularization strength. To address this issue, we introduce DAVA, a novel training procedure for variational auto-encoders. DAVA completely alleviates the problem of hyperparameter selection. We compare DAVA to models with optimal hyperparameters. Without any hyperparameter tuning, DAVA is competitive on a diverse range of commonly used datasets. Underlying DAVA, we discover a necessary condition for unsupervised disentanglement, which we call PIPE. We demonstrate the ability of PIPE to positively predict the performance of downstream models in abstract reasoning. We also thoroughly investigate correlations with existing supervised and unsupervised metrics. The code is available at github.com/besterma/dava.

## 1 Introduction

Real-world data tends to be highly structured, full of symmetries and patterns. This implies that there exists a lower-dimensional set of ground truth factors that is able to explain a significant portion of the variation present in real-world data. The goal of disentanglement learning is to recover these factors, so that changes in a single ground truth factor are reflected only in a single latent dimension of a model (see Figure 1 for an example). Such an abstraction allows for more efficient reasoning (Van Steenkiste et al., 2019) and improved interpretability (Higgins et al., 2017a). It further shows positive effects on zero-shot domain adaption (Higgins et al., 2017b) and data efficiency (Duan et al., 2020; Schott et al., 2022).

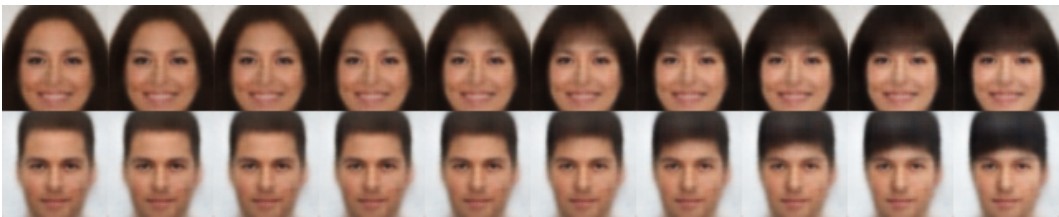

Figure 1: Latent traversals of a single latent dimension (hair fringes) of DAVA trained on *CelebA*. DAVA visibly disentangles the fringes from all other facial properties.

If the generative ground-truth factors are known and labeled data is available, one can train a model in a supervised manner to extract the ground-truth factors. What if the generative factors are unknown, but one still wants to profit from the aforementioned benefits for a downstream task? This may be necessary when the amount of labeled data for the downstream task is limited or training is computationally expensive. Learning disentangled representations in an unsupervised fashion is generally impossible without the use of some priors (Locatello et al., 2019b). These priors can be present both implicitly in the model architecture and explicitly in the loss function (Tschannen et al., 2018). An example of such a prior present in the loss function is a low total correlation between

latent variables of a model (Chen et al., 2018; Kim & Mnih, 2018). Reducing the total correlation has been shown to have a positive effect on disentanglement (Locatello et al., 2019b). Unfortunately, as we show in more detail in this work, how much the total correlation should be reduced to achieve good disentanglement is highly dataset-specific. The optimal hyperparameter setting for one dataset may yield poor results on another dataset. To optimize regularization strength, we need a way to evaluate disentanglement quality.

So how can we identify well disentangled representations? Evaluating representation quality, even given labeled data, is no easy task. Perhaps as an example of unfortunate nomenclature, the often-used term "ground-truth factor" implies the existence of a canonical set of orthogonal factors. However, there are often multiple equally valid sets of ground truth factors, such as affine transformations of coordinate axes spanning a space, different color representations, or various levels of abstraction for group properties. This poses a problem for supervised disentanglement metrics, since they fix the ground truth factors for evaluating a representation and judge the models too harshly if they have learned another equally valid representation. Furthermore, acquiring labeled data in a practical setting is usually a costly endeavor. The above reasons hinder the usability of supervised metrics for model selection.

In this work, we overcome these limitations for both learning and evaluating disentangled representations. Our improvements are based on the following idea: We define two distributions that can be generated by a VAE. Quantifying the distance between these two distributions yields a disentanglement metric that is independent of the specific choice of ground truth factors and reconstruction quality. The further away these two distributions are, the less disentangled the VAEs latent space is. We show that the similarity of the two distributions is a necessary condition for disentanglement. Furthermore, we can exploit this property at training time by introducing an adversarial loss into classical training of VAEs. To do this, we introduce a discriminator network into training and use the VAEs decoder as generator. During training, we control the weight of the adversarial loss. We adjust the capacity of the latent space information bottleneck accordingly, inspired by (Burgess et al., 2017). In this way, we allow the model to increase the complexity of its representation as long as it is able to disentangle.

This dynamic training procedure solves the problem of dataset-specific hyperparameters and allows our approach to reach competitive disentanglement on a variety of commonly used datasets without hyperparameter tuning.

Our contributions are as follows:

- We identify a novel unsupervised aspect of disentanglement called PIPE and demonstrate its usefulness in a metric with correlation to supervised disentanglement metrics as well as a downstream task.
- We propose an adaptive adversarial training procedure (DAVA) for variational auto-encoders, which solves the common problem that disentanglement performance is highly dependent on dataset-specific regularization strength.
- We provide extensive evaluations on several commonly used disentanglement datasets to support our claims.

## 2 RELATED WORK

### 2.1 MODEL ARCHITECTURES

The $\beta$-VAE by Higgins et al. (2017a) is a cornerstone model architecture for disentanglement learning. The loss function of the $\beta$-VAE, the evidence lower bound (ELBO), consists of a reconstruction term and a KL-divergence term weighted by $\beta$, which forces the aggregated posterior latent distribution to closely match the prior distribution. The KL-divergence term seems to promote disentanglement as shown in (Rolinek et al., 2019). The $\beta$-TCVAE architecture proposed by Chen et al. (2018) further decomposes the KL divergence term of the ELBO into an index-code mutual information, a total correlation and a dimension-wise KL term. They are able to show that it is indeed the total correlation that encourages disentanglement and propose a tractable but biased Monte Carlo estimate. Similarly, the FactorVAE architecture (Kim & Mnih, 2018) uses the density ratio trick with an adversarial network to estimate total correlation. The AnnealedVAE architecture (Burgess et al., 2017) as well

as ControlVAE (Shao et al., 2020) build on a different premise, arguing that slowly increasing the information bottleneck capacity of the latent space leads to the model gradually learning new latent dimensions and thus disentangling them. We will use a similar but optimized approach for DAVA. More recent promising approaches are presented by Wei et al. (2021) using orthogonal Jacobian regularization and by Chen et al. (2021) applying regulatory inductive bias recursively over a compositional feature space.

## 2.2 Introduction of Adversarial Training to VAEs

When combining autoencoders (AEs) with an adversarial setting, one can connect the adversarial network either on the latent space or on the output of the decoder. Makhzani et al. (2016) proposed an adversarial AE (AAAE) that uses an adversarial discriminator network on the latent space of an AE to match its aggregated posterior to an arbitrary prior. There, the encoder of the AE acts as the generator of the GAN and is the only part of the AE that gets updated with respect to the discriminator's loss. This kind of training is strongly connected to VAE training, with the adversarial loss taking on the role of KL divergence in the classical VAE training objective but without the constraint of a Gaussian prior. The previously mentioned FactorVAE (Kim & Mnih, 2018) implements an adversarial loss on the latent space to reduce total correlation. Larsen et al. (2016) proposed using a discriminator network with the decoder of the VAE acting as generator to improve the visual fidelity of the generated images, but with no focus on disentanglement. The difference to DAVA is that they used the discriminator on the real observations, while we propose a discriminator that only sees observations generated by the decoder of the VAE. Zhu et al. (2020) introduce an recognition network based loss on the decoder that encourages predictable latent traversals. In other words, given a pair of images where all but one latent dimensions are kept constant, the recognition network should be able to predict in which latent dimension the change occurred. Applied on top of a baseline VAE model, this loss slightly improves the disentanglement performance of the baseline VAE. In a semi-supervised setting, Carbajal et al. (2021) and Han et al. (2020) propose adversarial losses on the latent space of VAEs to disentangle certain parts of the latent space from information present in labels. Unfortunately, such an approach does not work in an unsupervised setting with the goal of disentanglement of the complete latent space.

## 2.3 Measuring Disentanglement

All supervised metrics have in common that they are dependent on a canonical factorization of ground truth factors. Given access to the generative process of a dataset, the FVAE metric (Kim & Mnih, 2018) can be used to evaluate a model. It first creates a batch of data generated by keeping aforementioned ground truth factor fixed and randomly sampling from the other ground truth factors. It then uses a majority vote classifier to predict the index of a ground truth factor given the variance of each latent dimension computed over said batch. Without access to the generative process, but given a number of fully labeled samples, one can use metrics like DCI Disentanglement by Eastwood & Williams (2018) and MIG by Chen et al. (2018). While DCI in essence assesses if a latent variable of a model captures only a single ground truth factor, MIG evaluates if a ground truth factor is captured only by a single variable. As a result, DCI compared to MIG does not punish multi-dimensional representations of a single ground truth factor, for example the RGB model of color or a sine/cosine representation of orientation.

There exists a small number of unsupervised disentanglement metrics. The unsupervised disentanglement ranking (UDR) by Duan et al. (Duan et al., 2020) evaluates disentanglement based on the assumption that a representation should be disentangled if many models, differently initialized but trained with the same hyperparameters, learn the same representation. To achieve this, they compute pairwise similarity (up to permutation and sign inverse) of the representations of a group of models. The score of a single model is the average of its similarity to all the other models in the group. ModelCentrality (MC) (Lin et al., 2020) builds on top of UDR by improving the pairwise similarity evaluation. A drawback is the high computational effort, as to find the optimal hyperparameter setting, multiple models need to be trained for each setting. UDR and MC do not assume any fixed set of ground truth factors. Nevertheless, a weakness of these approaches is that they do not recognize similarity of a group of models that each learn a different bijective mapping of the ground truth factors. The latent variation predictability (VP) metric by Zhu et al. (2020) is based on the assumption that if a representation is well disentangled, it should be easy for a recognition network to predict

which variable was changed given two input images with a change in only one dimension of the representation. An advantage of the VP metric is its ability to evaluate GANs as it is not dependent on the model containing an encoder. In comparison to our proposed metric, VP, UDR and MC are dependent on the size of the latent space or need to define additional hyperparameters to recognize inactive dimensions. Estermann et al. (2020) showed that the UDR has a strong positive bias for low-dimensional representations, as low-dimensional representations are more likely to be similar to each other. One could see the same issue arise for the VP metric, as accuracy for the recognition network will likely be higher when there is only a low number of change-inducing latent dimensions available to choose from.

### 2.4 APPLICATIONS OF DISENTANGLED REPRESENTATIONS

The effects of disentangled representations on downstream task performance are manifold. Van Steenkiste et al. (2019) showed that for abstract reasoning tasks, disentangled representations enabled quicker learning. We show in Section 4.2 that the same holds for our proposed metric. In work by Higgins et al. (2017b), disentangled representations provided improved zero-shot domain adaption for a multi-stage reinforcement-learning (RL) agent. The UDR metric (Duan et al., 2020) correlated well with data efficiency of a model based RL agent introduced by Watters et al. (Watters et al., 2019). Disentangled representations further seem to be helpful in increasing the fairness of downstream prediction tasks (Locatello et al., 2019a). Contrary to previous qualitative evidence (Eslami et al., 2018; Higgins et al., 2018), Montero et al. (2020) present quantitative findings indicating that disentanglement does not have an effect on out-of-distribution (OOD) generalization. Recent work by Schott et al. (2022) supports the claims of Montero et al. (2020), concluding that disentanglement shows improvement in downstream task performance but not so in OOD generalization.

## 3 POSTERIOR INDIFFERENCE PROJECTION EQUIVALENCE

We first introduce the notation of the VAE framework, closely following the notation used by Kim & Mnih (2018). We assume that observations $x \sim \tilde{D}$ are generated by an unknown process based on some independent ground-truth factors. The goal of the encoder is to represent $x$ in a latent vector $z \in \mathbb{R}^d$. We introduce a Gaussian prior $p(z) = \mathcal{N}(0, I)$ with identity covariance matrix on the latent space. The variational posterior for a given observation $x$ is then $q_\theta(z|x) = \prod_{j=1}^d \mathcal{N}(z_j|\mu_j(x), \sigma_j^2(x))$, where the encoder with weights $\theta$ outputs mean and variance. The decoder with weights $\phi$ projects from the latent space $z$ back to observation space $p_\phi(x|z)$. We can now define the distribution $q(z)$.

**Definition 3.1** (EP)**.** The empirical posterior (EP) distribution $q(z)$ of a VAE is the multivariate distribution of the latent vectors $z$ over the data distribution $\tilde{D}$. More formally, $q(z) = \mathbb{E}_{x \sim \tilde{D}}[q_\theta(z|x)]$. We can reconstruct an observation $x \sim \tilde{D}$ the following way: We sample $z \sim q_\theta(z|x)$ and then get the reconstruction $\hat{x} \sim p_\phi(x|z)$. We informally call observations generated by this process reconstructed samples and denote them as $\hat{x}$.

The decoder is not constrained to only project from $q(z)$ to observation space. We can sample observations from the decoder by using different distributions on $z$. We define a particularly useful distribution.

**Definition 3.2** (FP)**.** The factorial posterior (FP) distribution $\bar{q}(z)$ of a VAE is a multivariate distribution with diagonal covariance matrix. We define it as the product of the marginal EP distribution: $\bar{q}(z) = \prod_{j=1}^d q(z_j)$. We can use the decoder to project $z \sim \bar{q}(z)$ to observation space $\tilde{x} \sim p_\phi(x|z)$. We informally call observations created by this process generated samples and denote them as $\tilde{x}$.

We can now define the data distributions that arise when using the decoder to project images from either the EP or the FP.

**Definition 3.3** ($\tilde{D}_{\text{EP}}, \tilde{D}_{\text{FP}}$)**.** $\tilde{D}_{\text{EP}}$ is generated by the decoder projecting observations from the EP latent distribution $q(z)$, i.e. reconstructed samples. $\tilde{D}_{\text{EP}} = \mathbb{E}_{z \sim q(z)}[p_\phi(x|z)]$. $\tilde{D}_{\text{FP}}$ is generated by the decoder projecting observations from the FP distribution $\bar{q}(z)$, i.e. generated samples. $\tilde{D}_{\text{FP}} = \mathbb{E}_{z \sim \bar{q}(z)}[p_\phi(x|z)]$.

We can now define the core concept of this paper. We look at the similarity of two data distributions generated by the decoder, $\tilde{D}_{\text{EP}}$ and $\tilde{D}_{\text{FP}}$.

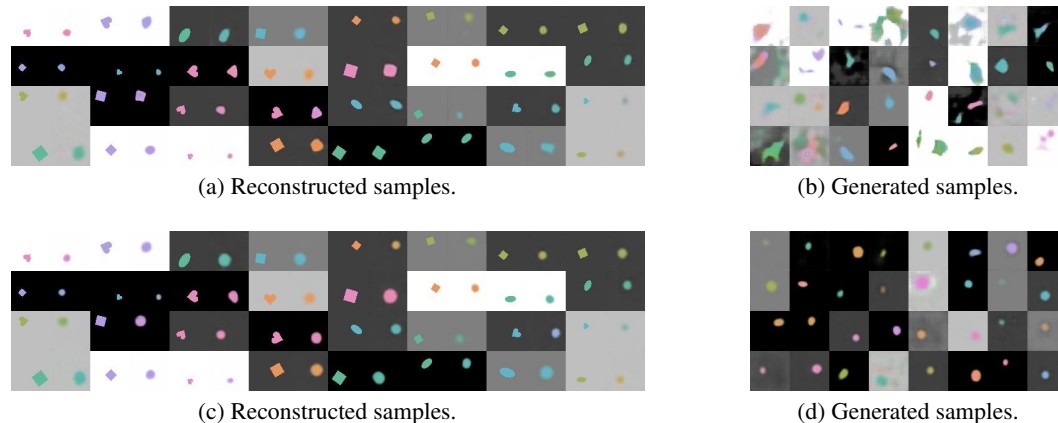

(a) Reconstructed samples.

(b) Generated samples.

(c) Reconstructed samples.

(d) Generated samples.

Figure 2: Shown are two models of `disentanglement-lib` (Locatello et al., 2019b) with different disentanglement performance. The upper part shows a poorly disentangled model while the lower part shows a comparatively well disentangled model. In (a) and (c), odd columns show images sampled from the *AbstractDSprites* dataset, even columns show the corresponding model reconstruction. (b) and (d) show images generated by the decoder of each model given the factorial posterior distribution. While the upper model achieves better reconstructions (a), its generated samples (b) are visibly out of distribution. The bottom model ignores shape in its reconstructions (c), but its generated samples (d) capture the true data distribution more accurately.

**Definition 3.4** (PIPE). The posterior indifference projection equivalence (PIPE) of a VAE represents the similarity of the data distributions $\tilde{D}_{\mathrm{EP}}$ and $\tilde{D}_{\mathrm{FP}}$. In other words, PIPE is a measure of the decoder's indifference to the latent distribution it projects from.
$PIPE(\theta, \phi) = \omega(\mathbb{E}_{z \sim q(z)}[p_\phi(x|z)], \mathbb{E}_{z \sim \bar{q}(z)}[p_\phi(x|z)])$, where $\omega$ is a general similarity measure.

Unsupervised disentanglement learning without any inductive biases is impossible as has been proven by Locatello et al. (2019b). Given a disentangled representation, it is always possible to find a bijective mapping that leads to an entangled representation. Without knowledge of the ground truth factors, it is impossible to distinguish between the two representations. We argue that PIPE is necessary for a disentangled model, even when it is not sufficient. Suppose a model has learned a disentangled representation, but $\tilde{D}_{\mathrm{EP}}$ is not equivalent to $\tilde{D}_{\mathrm{FP}}$. There are two cases where this could happen. The first possibility is that the latent dimensions of the model are not independent. This violates the independence assumption of the ground truth factors, so the model cannot be disentangled. The second possibility is that the model has not learned a representation of the ground truth factors and is generating samples that are not represented by the ground truth factors. This model would not be disentangled either. We conclude that PIPE is a necessary condition for disentanglement.

In Figure 2 we show reconstructed and generated samples of two different models to support our chain of reasoning. Generated samples of the entangled model (Figure 2 (b)) look visibly out of distribution compared to the reconstructed samples. Generated samples of the disentangled model (Figure 2 (d)) appear to be equivalent to the distribution of the reconstructed samples.

## 4 PIPE METRIC

We propose a way to quantify the similarity metric $\omega$ of PIPE with a neural network and call this the PIPE metric. Given a model $\mathcal{M}$ with encoder weights $\theta$, decoder weights $\phi$, and corresponding $\tilde{D}_{\mathrm{EP}}, \tilde{D}_{\mathrm{FP}}$ (see Appendix B.2 for details on how to sample from $\tilde{D}_{\mathrm{EP}}$ and $tildeD_{FP}$).

1. Create a set of observations $S_{EP}$ by sampling from $\tilde{D}_{\mathrm{EP}}$. Informally, these are the reconstructed samples.

2. Create a set of observations $S_{FP}$ by sampling from $\tilde{D}_{\mathrm{FP}}$. Informally, these are the generated samples.

3. Randomly divide $S_{EP} \cup S_{FP}$ into a train and a test set.

4. Train a discriminator network on the train set.

5. Evaluate accuracy $acc$ of the discriminator on the test set.

6. Since a random discriminator will guess half of the samples accurately, we report a score of $2 \cdot (1 - acc)$, such that 1 is the best and 0 is the worst score.

We train the discriminator network for $10,000$ steps to keep the distinction of $S_{EP}$ and $S_{FP}$ sufficiently difficult. We further use a uniform factorial distribution instead of FP for a slight improvement in performance. Details on the implementation can be found in Appendix B.

### 4.1 Relation to Existing Metrics

To classify the performance of the PIPE metric, we evaluate correlations with supervised metrics on a diverse set of commonly used datasets. We namely consider *Shapes3D* (Burgess & Kim, 2018), *AbstractDSprites* (Van Steenkiste et al., 2019) and *Mpi3d Toy* (Gondal et al., 2019). All datasets are part of `disentanglement-lib` (Locatello et al., 2019b), which we used to train the models we evaluated our metric on. More details on the implementation of the PIPE metric and the evaluated models can be found in Appendix B. Results are displayed in Figure 3. They show that the PIPE metric correlates positively with existing supervised metrics DCI, MIG and FVAE, surpassing the correlations of the unsupervised baselines. While the performance of UDR on *Mpi3d Toy* and the performance of MC on *AbstractDSprites* is lacking, PIPE demonstrates consistent performance across all datasets. Further, as opposed to UDR and MC, PIPE can be evaluated on a single model only, whereas UDR and MC are only able to evaluate sets of models.

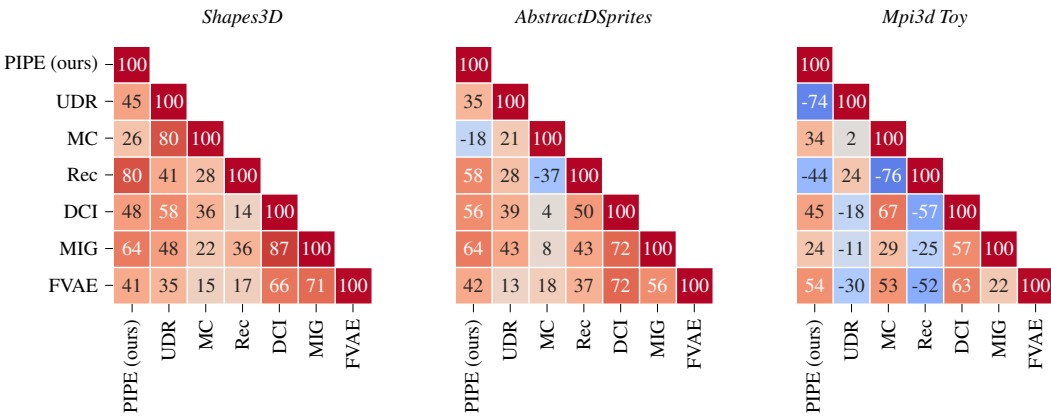

Figure 3: Spearman rank correlation between different metrics on three different datasets. Correlations take values in the range $[-1, 1]$, where 1 means perfect correlation, 0 means no correlation, and negative values mean anti-correlation. For better readability, we have multiplied all values by 100. PIPE, UDR (Duan et al., 2020) and MC (Lin et al., 2020) are the corresponding unsupervised metrics. We include Reconstruction Loss Rec as a trivial unsupervised baseline. DCI, MIG and FVAE are the corresponding supervised metrics. Low or even negative correlations of our metric with UDR and MC show that our metric captures a different aspect of disentanglement. The correlation of our metric with supervised disentanglement metrics is mostly consistent across datasets. We note that the correlation with supervised metrics on the *Mpi3d Toy* dataset changes direction for both UDR and Rec, while MC has difficulties with *AbstractDSprites*.

### 4.2 Correlation with Downstream Task Performance

To further quantify the usefulness of the PIPE metric, we analyze the predictive performance of the metric for an abstract reasoning downstream task. For the downstream task, a VAE is trained to learn a disentangled representation of a dataset. The downstream model then only gets access to this representation then trying to solve the downstream task. Van Steenkiste et al. (2019) evaluated

different supervised disentanglement metrics in terms of predictive performance of accuracy in an abstract reasoning task on the datasets *Shapes3D* and *AbstractDSprites*. They showed that good disentanglement is an indicator of better accuracy during the early stages on training. We reproduce their experiment on a reduced scale by only considering up to 6,000 training steps. As can be seen in Figure 4, our metric is on par with supervised metrics and clearly outperforms the unsupervised baselines. More importantly this means that PIPE is a positive predictor of accuracy of downstream models in a few-sample setting and is therefore a desirable property for unsupervised model selection.

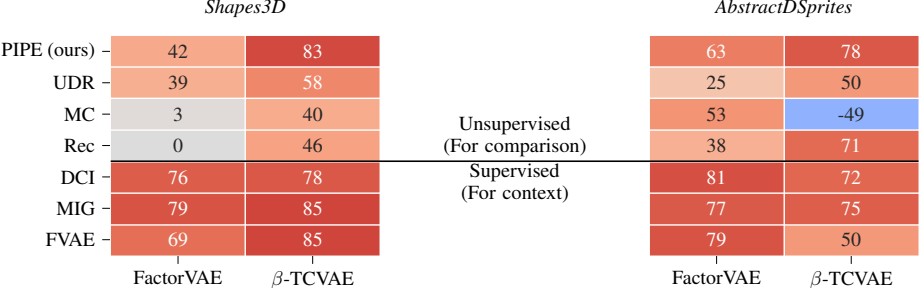

Figure 4: Spearman rank correlation between different disentanglement metrics and downstream accuracy of the abstract visual reasoning task (Van Steenkiste et al., 2019) after 6,000 training steps for FactorVAE and $\beta$-TCVAE. The same metrics as in Figure 3 are evaluated. Correlation for unsupervised metrics are more sensitive to model architecture than for supervised metrics. PIPE shows higher correlation with downstream performance than UDR and clearly outperforms the Rec baseline.

## 5    TRAINING PROCEDURE FOR VARIATIONAL AUTO-ENCODERS (DAVA)

In the previous chapter we have established that PIPE is desirable for a model. We now present how to encourage this property at training time by designing an adversarial training procedure. We train a discriminator and a VAE at the same time. The discriminator needs to differentiate $\tilde{D}_{\text{EP}}$ from $\tilde{D}_{\text{FP}}$, which is achieved with the following loss:

$$\mathcal{L}_{dis} = \mathbb{E}_{\hat{x} \sim \tilde{D}_{\text{EP}}}[\log(Dis(\hat{x})] + \\ \mathbb{E}_{\tilde{x} \sim \tilde{D}_{\text{FP}}}[\log(1 - Dis(\tilde{x}))].$$

The loss function for the VAE is compromised of a reconstruction term with the weighted negative objective of the discriminator:

$$\mathcal{L}_{adv} = \mathcal{L}_{rec} - \mu \mathcal{L}_{dis}.$$

with reconstruction loss

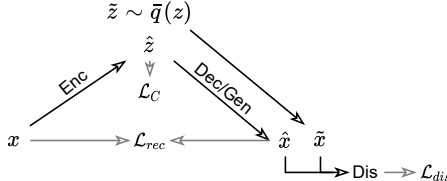

Figure 5: Overview of the composition of DAVA. A discriminator attached to the decoder of a VAE. The goal for the discriminator is to differentiate samples coming from the EP from those coming from the FP.

$$\mathcal{L}_{rec} = \mathbb{E}_{q_\theta(z|x)}[\log p_\phi(x|z)]$$

While this works in practice, it faces the issue of the weight $\mu$ being very dataset-specific. It is also closely related to FactorVAE, with the difference that FactorVAE applies the adversarial loss to the latent space and to the encoder only. Let us consider an alternative approach. Work by Burgess et al. (2017) looked at the KL-Divergence from an information bottleneck perspective. They proposed using a controlled latent capacity increase by introducing a loss term on the deviation of some goal capacity $C$ that increases during training:

$$\mathcal{L}_C = |\text{KL}(q(z|x)||p(z)) - C|$$

Such a loss provides a disentanglement prior complementary to the minimizing total correlation prior used in $\beta$-TCVAE and FactorVAE. Unfortunately, it too depends on specific choice of hyperparameters, mainly the max capacity $C$ and the speed of capacity increase. We now demonstrate how to

incorporate both the adversarial loss and the controlled capacity increase into a single loss function. We call this approach disentangling adversarial variational auto-encoder DAVA (also see Figure 5):

$$\mathcal{L}_{DAVA} = \mathcal{L}_{rec} - \gamma\mathcal{L}_C - \mu\mathcal{L}_{dis}$$

The main strength of DAVA is its ability to dynamically tune $C$ and $\mu$ by using the accuracy of its discriminator. This yields a model that performs well on a diverse range of datasets. We now outline the motivation and the method for tuning $C$ and $\mu$ during training.

The goal of the controlled capacity increase in the AnnealedVAE architecture is to allow the model to learn individual factors at a time into individual latent dimensions. The order in which they are learned corresponds to their respective contribution to the reconstruction loss. In DAVA, we want to encourage the model to *learn new factors by increasing $C$ as long as it has learned a disentangled representation of the currently learned factors*. This is the case when the discriminator cannot distinguish $\tilde{D}_{EP}$ from $\tilde{D}_{FP}$. As soon as the accuracy of the discriminator increases, we want to stop increasing $C$ and increase the weight of the adversarial loss to ensure that the model does not learn any new factors of variation while helping it to disentangle the ones it has picked up into individual dimensions. As can be seen in Figure 6, this results in dataset-specific schedules of $C$ over the course of the training. Algorithm 1 in Appendix C.1 describes the training of DAVA in more detail.

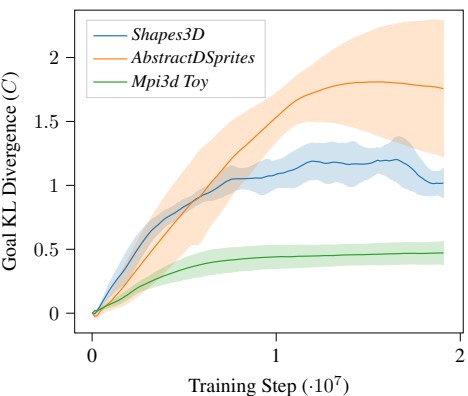

Figure 6: Information bottleneck capacity $C$ during training of DAVA for different datasets. Depending on the complexity and structure of the dataset as well as random initialization of the model, a different schedule of $C$ is necessary to achieve best performance. The shaded area denotes one standard deviation across 5 random seeds.

## 5.1 EXPERIMENTS

To quantify the performance of DAVA, we perform an extensive study including multiple model architectures and datasets. As a baseline, we include the best disentangling models according to (Locatello et al., 2019b), namely the $\beta$-TCVAE (Chen et al., 2018) and FactorVAE (Kim & Mnih, 2018). We also include AnnealedVAE (Burgess et al., 2017) and ControlVAE (Shao et al., 2020). As discussed earlier, the performance of each of these approaches is highly dependent on the regularization strength. For this reason, we perform a grid-search over a range of values curated by roughly following the work of Locatello et al. (2019b) and adjusting where necessary to ensure best possible performance. The exact hyperparameters considered can be found in appendix C.1. The training of DAVA does not rely on such dataset-specific regularization, therefore **all hyperparameters** for the training of DAVA were kept **constant** across **all** considered datasets. All other hyperparameters closely follow (Locatello et al., 2019b) and were kept consistent across all architectures (baseline and DAVA) and are also reported in appendix C.1. We run 5 different random seeds for each configuration to determine statistical significance. We provide summarized results for the most informative datasets in Tables 1, 2 and 3. For the baselines we report the performance of the mean hyperparameter setting according to the average score of all considered metrics. The reader is advised to consult Appendix D for complete results of all evaluated datasets.

Table 1: Results *Shapes3D*

| Architecture | MIG | DCI | FVAE | PIPE |
|---|---|---|---|---|
| $\beta$-TCVAE | 0.26±0.06 | 0.50±0.03 | 0.76±0.03 | 0.11±0.03 |
| FactorVAE | 0.20±0.10 | 0.41±0.07 | 0.78±0.04 | 0.22±0.04 |
| AnnealedVAE | 0.52±0.02 | 0.70±0.02 | **0.92±0.01** | 0.26±0.03 |
| ControlVAE | 0.07±0.03 | 0.17±0.04 | 0.68±0.07 | 0.02±0.00 |
| Ours | **0.62±0.05** | **0.78±0.03** | 0.82±0.03 | **0.61±0.04** |

Table 2: Results *AbstractDSprites*

| Architecture | MIG | DCI | FVAE | PIPE |
|---|---|---|---|---|
| $\beta$-TCVAE | 0.12±0.01 | 0.18±0.01 | 0.45±0.02 | 0.19±0.01 |
| FactorVAE | 0.12±0.02 | 0.19±0.02 | 0.55±0.03 | 0.20±0.03 |
| AnnealedVAE | 0.15±0.01 | 0.21±0.01 | 0.54±0.02 | 0.18±0.01 |
| ControlVAE | 0.04±0.02 | 0.07±0.01 | 0.42±0.02 | 0.06±0.01 |
| Ours | **0.23±0.04** | **0.27±0.05** | **0.67±0.05** | **0.35±0.03** |

Table 3: Results *Mpi3d Toy*

| Architecture | MIG | DCI | FVAE | PIPE |
|---|---|---|---|---|
| $\beta$-TCVAE | 0.11±0.02 | 0.23±0.01 | 0.39±0.02 | 0.09±0.01 |
| FactorVAE | 0.02±0.01 | 0.13±0.00 | 0.38±0.02 | (0.99±0.10) |
| AnnealedVAE | 0.07±0.03 | 0.23±0.02 | **0.50±0.02** | 0.02±0.01 |
| ControlVAE | 0.04±0.01 | 0.17±0.02 | 0.43±0.04 | 0.03±0.01 |
| Ours | **0.12±0.09** | **0.30±0.03** | 0.41±0.04 | **0.21±0.03** |

Appendix D also contains the results of the best performing hyperparameter choice for each baseline model, demonstrating their variance across datasets. Note that the PIPE scores of FactorVAE in Table 3 are exploiting a design decision discussed in depth in Appendix B.3. The decoders of these models mapped all latent values to a single image, therefore we exclude their scores. The results showcase the strong consistent performance of DAVA across different datasets. Compared to the baselines, DAVA is able to deliver improvements of up to 50% for the supervised metrics, and over 100% for PIPE. This allows DAVA to be used in practice as consistent performance can be expected.

We further validate DAVA on three real-world datasets. CelebA (Liu et al., 2015) is dataset commonly used in the disentanglement literature. It compromises of over 200,000 images of faces of celebrities. As shown in Figure 7, DAVA achieves a competitive FID while still having a comparatively high PIPE metric score. This means that DAVA accomplishes a similar visual fidelity of its generated samples as the best baseline methods while better disentangling its latent space. Results for datasets *Mpi3d Real* (Gondal et al., 2019) and *CompCars* (Yang et al., 2015) can be found in Appendix D and E, respectively.

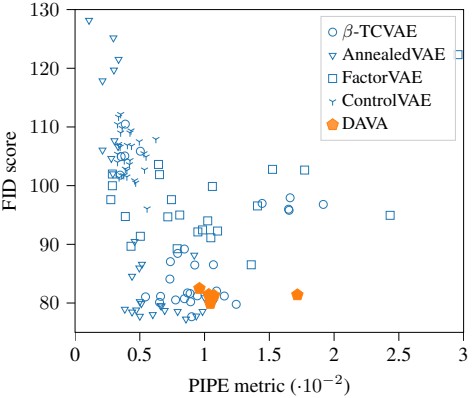

## 6 CONCLUSIONS

In this work, we presented a new way to understand and formalize disentangled representations from an unsupervised point of view. The main idea is that the distribution of reconstructed samples of a disentangled VAE should be similar to the distribution of generated samples. We quantify this notion in a new unsupervised disentanglement metric called PIPE metric for evaluating disentangled representations. Without access to ground truth, PIPE provides results consistent with supervised metrics. Furthermore,

Figure 7: The FID score (Heusel et al., 2017) measures the distance between two different distributions of images and is commonly used in GAN literature to evaluate the quality of the generator. A low FID indicates that a model generates visually convincing images. We evaluate the FID of image samples from $\tilde{D}_{\text{FP}}$ of the respective models against the original images. We find that DAVA achieves a competitive FID with comparatively good PIPE. We have removed a small number of poorly performing outliers from the baseline models to improve readability.

we apply the same idea to DAVA, a novel VAE training procedure. DAVA can self-tune its regularization strength and produces results competitive with existing models that require more dataset-specific hyperparameter tuning. Robust unsupervised disentanglement performance is an important requirement for practical applicability. With DAVA, we provide a method that can be used by practitioners with relatively low computational requirements.

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

# A  DATASETS

We provide a short summary of each dataset, including samples and the respective data generating ground truth factors.

## A.1  SHAPES3D

The *Shapes3D* dataset (Burgess & Kim, 2018) includes the following ground truth factors: floor hue, wall hue, object hue, scale, shape and orientation.

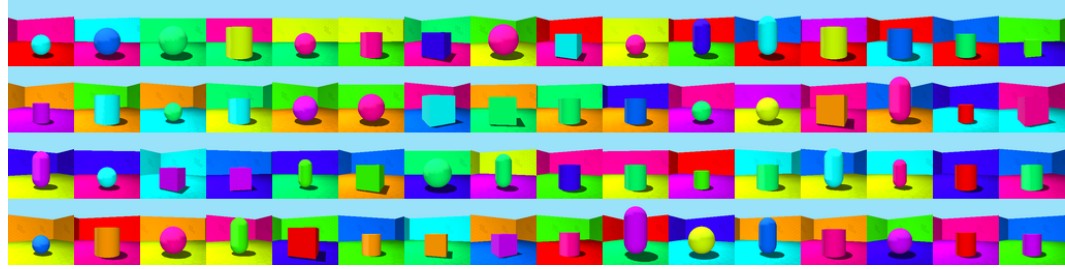

Figure 8: Samples from *Shapes3D*.

## A.2  DSPRITES

The *DSprites* dataset (Matthey et al., 2017) includes the following ground truth factors: shape, scale, orientation, x-position and y-position.

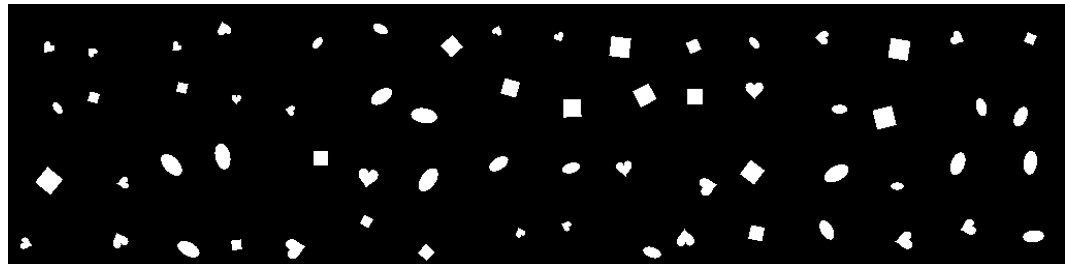

Figure 9: Samples from *DSprites*.

## A.3  ABSTRACTDSPRITES

The *AbstractDSprites* dataset (Van Steenkiste et al., 2019) includes the following ground truth factors: background color, object color, shape, scale, x-position and y-position.

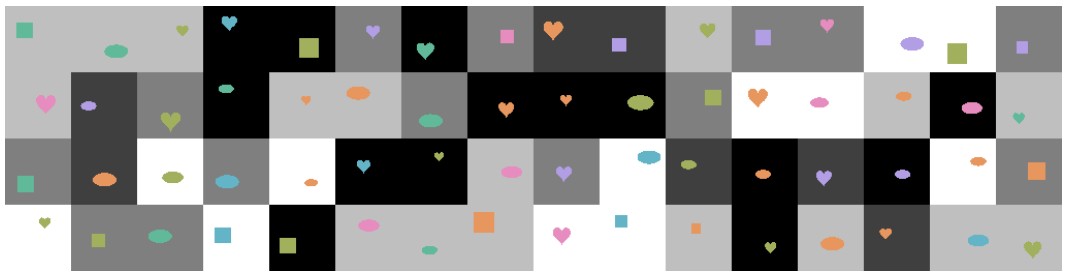

Figure 10: Samples from *AbstractDSprites*.

### A.4 MPI3D TOY

The *Mpi3d Toy* dataset (Gondal et al., 2019) includes the following ground truth factors: object color, object shape, object size, camera height, background color, first degree of freedom, second degree of freedom. It consists of low-quality renders of the experimental setup.

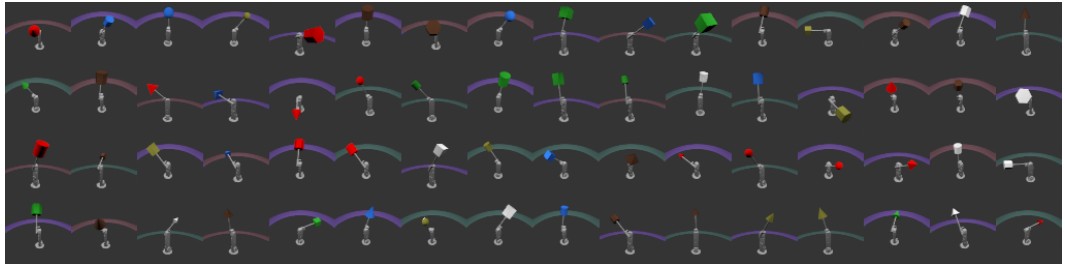

Figure 11: Samples from *Mpi3d Toy*.

### A.5 MPI3D REAL

The *Mpi3d Real* dataset (Gondal et al., 2019) includes the same ground truth factors as *Mpi3d Toy*. It consists of real pictures of an experimental setup.

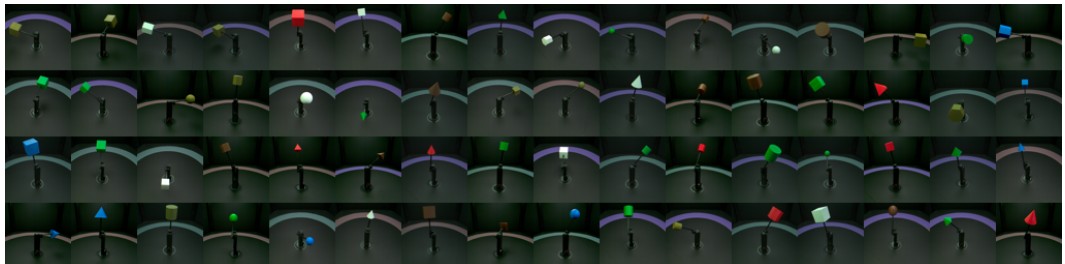

Figure 12: Samples from *Mpi3d Real*

### A.6 SMALLNORB

The *Smallnorb* dataset (LeCun et al., 2004) includes the following ground truth factors: object type, elevation, azimuth, lighting condition.

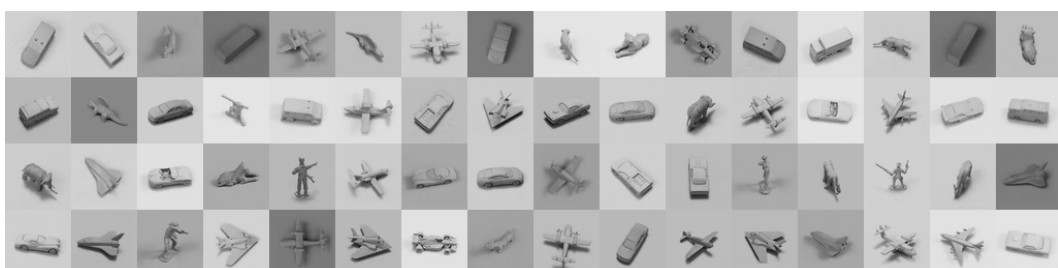

Figure 13: Samples from *Smallnorb*.

### A.7 NOISYDSPRITES

The *NoisyDSprites* dataset (Locatello et al., 2019b) includes the same ground truth factors as *DSprites*. Instead of the background being black, it consists of random noise.

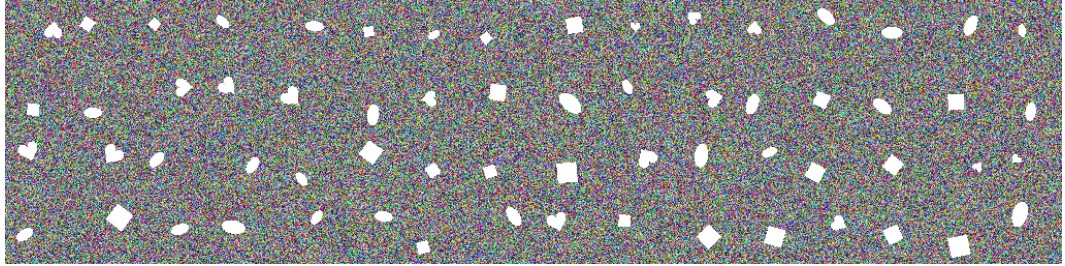

Figure 14: Samples from *NoisyDSprites*

## A.8    CARS3D

The *Cars3D* dataset (Reed et al., 2015) includes the following ground truth factors: elevation, azimuth, object type.

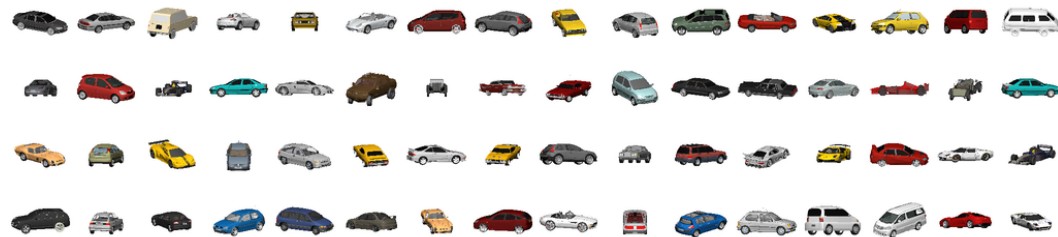

Figure 15: Samples from *Cars3D*.

## A.9    CELEBA

The *CelebA* dataset (Liu et al., 2015) includes multiple images of over 10.000 identities with a rich set of facial properties, poses and background clutter.

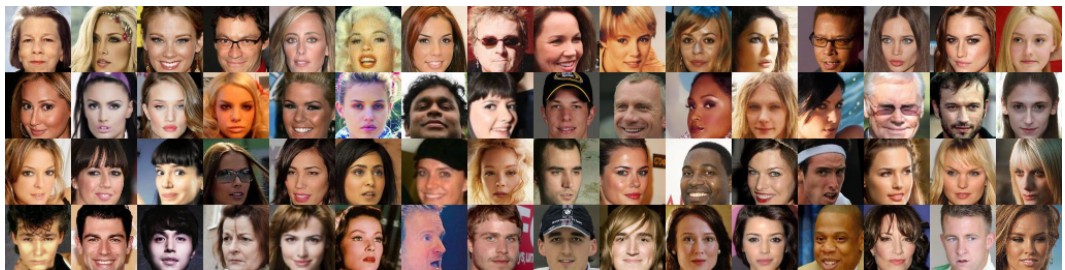

Figure 16: Samples from *CelebA*.

### A.10 COMPCARS

The *CompCars* dataset (Yang et al., 2015) contains images of 163 car makes with 1.716 car models, totalling to 136.726 images. The images scraped from the web and are composed of the car in different poses, eg. front and side view, with diverse backgrounds. It is uncleaned and might contain occlusions and highly correlated ground truth factors.

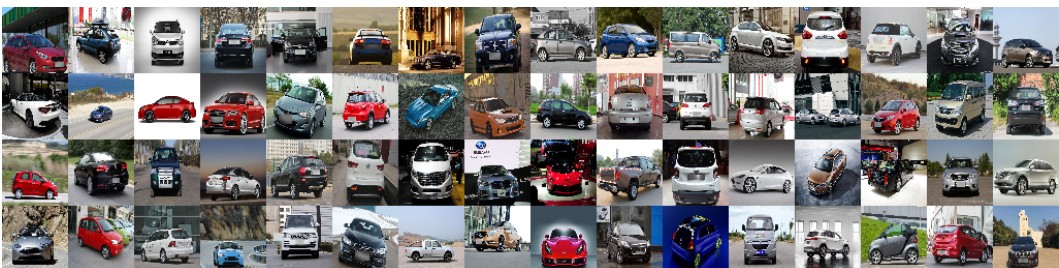

Figure 17: Samples from *CompCars*.

## B PIPE METRIC DETAILS

### B.1 TRAINING OF EVALUATED MODELS

We used `disentanglement-lib` (Locatello et al., 2019b), enabling easy reproducibility of our results, to train all models for *Shapes3D*, *AbstractDSprites* and *Mpi3d Toy*. We focus on FactorVAE and $\beta$-TCVAE as they are the models generally achieving the highest disentanglement performance in (Locatello et al., 2019b). Following (Locatello et al., 2019b) on a reduced scale, we trained 5 random seeds each for 6 different hyperparameters. This leads to a total of 60 model instances per dataset, spanning a broad range of disentanglement performance.

We followed a similar approach to reproduce the 30 FactorVAE and 30 $\beta$-TCVAE models for each dataset of the abstract reasoning study (Van Steenkiste et al., 2019). For each VAE model instance, we then trained 5 downstream models on the abstract reasoning task, based on the representation of the VAE, for 6000 steps and evaluated their performance. We used `disentanglement-lib` to compute the metrics for all evaluated models.

### B.2 SAMPLING STRATEGIES FOR $\tilde{D}_{\text{EP}}$ AND $\tilde{D}_{\text{FP}}$

To sample from $\tilde{D}_{\text{EP}}$, one can simply sample random instances from the data distribution and then use the model $\mathcal{M}$ to reconstruct these instances. This implicitly computes the EP latent distribution $q(z)$. To sample from $S_{FP}$, we first need to estimated $\bar{q}(z)$. Closely following Algorithm 1 in (Kim & Mnih, 2018), we can achieve this by sampling a batch from $q(z)$ and then randomly permuting across the batch for each latent dimension (see *PermuteDims* in Algorithm 1). This is a standard trick used in the independence testing literature (Arcones & Gine, 1992). As long as the batch is large enough, the distribution of these samples samples will closely approximate $\bar{q}(z)$. For the evaluation of the PIPE metric, it is therefore not necessary to compute the marginals over the complete data distribution. To amplify the differences between $S_{EP}$ and $S_{FP}$, we then slightly adjusted the sampling strategy for the factorial posterior (FP). For each dimension of the latent representation we first estimate its range while encoding a batch of observations. We then sample values uniformly in that range, separately for each dimension, resulting in a similar factorial distribution as described earlier, but including a dimension-wise uniform distribution prior. This slightly improves performance of the PIPE metric. The architecture of the discriminator used in PIPE is similar to the encoder of the VAE displayed in Table 4.

### B.3 THE ROLE OF RECONSTRUCTION LOSS

Consider a model which does not store any information in its latent space, where the decoder outputs the same image regardless of the inputs it receives. As the PIPE metric does not (and cannot) have

any knowledge of the ground truth factors, it will give such a model a perfect score[1]. It is possible to use the reconstruction loss as a proxy for how good a model manages to capture the ground truth factors. However, the usefulness of the reconstruction loss is limited by two factors. First, it is strongly affected by noise, which requires at least dataset-specific normalization for it to be used in a metric. Second, the contribution of different ground truth factors in the reconstruction loss can differ a lot. We define

$$PIPE_{Rec} = PIPE - \alpha \cdot Rec_{norm}$$

with $Rec_{norm}$ denoting the per dataset normalized reconstruction loss. We show results for $\alpha = 0.5$ and $\alpha = 1$ in Figure 18. While adding the reconstruction loss brings improved correlations with supervised metrics for *Shapes3D* and *Mpi3d Toy*, it decreases correlations for *AbstractDSprites*, especially for $\alpha = 1$. For this reason, we decided to propose PIPE without any reconstruction loss component, as a practitioner, depending on the task, might have different requirements. In practice, we suggest to use a classical (denoising) VAE to get a feeling for the achievable reconstruction loss on a dataset and then use PIPE to select the best disentangling models among those that achieve a satisfactory reconstruction loss.

### B.4    RELATION TO FID

The Fréchet inception distance (FID) (Heusel et al., 2017) is a metric to quantify the similarity between two image distributions. This is commonly achieved by using a pretrained Inception v3 model (Szegedy et al., 2016) without its final classification layer to encode the image distributions. The FID is then the Fréchet distance of two Gaussian distributions fitted to the respective latent image distributions. It is commonly used to evaluate the performance of GANs where one desires the generated images to be as similar as possible to the original dataset. While it appears that there are strong similarities between FID and PIPE, there exist distinct differences. The contribution of PIPE is to show the importance of the concept of similarity of the $D_{EP}$ and $D_{FP}$ distributions in light of disentanglement. Compared to our proposed PIPE metric, FID would be another way to quantify PIPE. Figure 19 shows that FID and the PIPE metric have some correlation, while also showing substantial differences for some models. A large advantage of the PIPE metric over FID would be that it is not dependent on a pretrained model. This opens up the possibility to deploy the PIPE metric on unseen data domains without any pretrained models available.

### B.5    COMPARISON TO MODELCENTRALITY

Even though MC outperforms UDR, PIPE still considerably outperforms MC. We discuss possible reasons on why this might be the case. To evaluate similarity of two models, MC samples from the prior of a model, but the EP distribution of the model might be far away from the prior. PIPE does not have this problem because it samples from the EP and FP distributions. To evaluate similarity of two models in MC, the second model is asked to encode generated samples of the first model (very similar to $\tilde{(D)}_{EP}$). This distribution might be far away from the initial data distribution the encoder of the second model is used to. Another issue is that if only a very small number of models actually finds a disentangled representation, it is hard for MC to identify these models. PIPE can identify even a single disentangled model among many. Further, due to implicit biases in model architecture, it is possible that models learn similar entangled representations, thus breaking the core assumption of MC that only disentangled representations are similar.

## C    HYPERPARAMETERS AND DAVA TRAINING DETAILS

### C.1    HYPERPARAMETERS

DAVA and all considered baseline models use the same VAE architecture as displayed in Table 4. The architecture of the discriminator and its specific hyperparameters used in FactorVAE are shown in Table 6. All models, DAVA and baseline, use the same general hyperparameters displayed in Table 5. We present the hyperparameter range explored for each baseline model in Table 7. We adjusted the latent space size z_dim for all models for a small number of datasets. We did this in order to give the

---

[1]In fact, this happened with a few FactorVAE models on *Mpi3d Toy*, see Table 3.

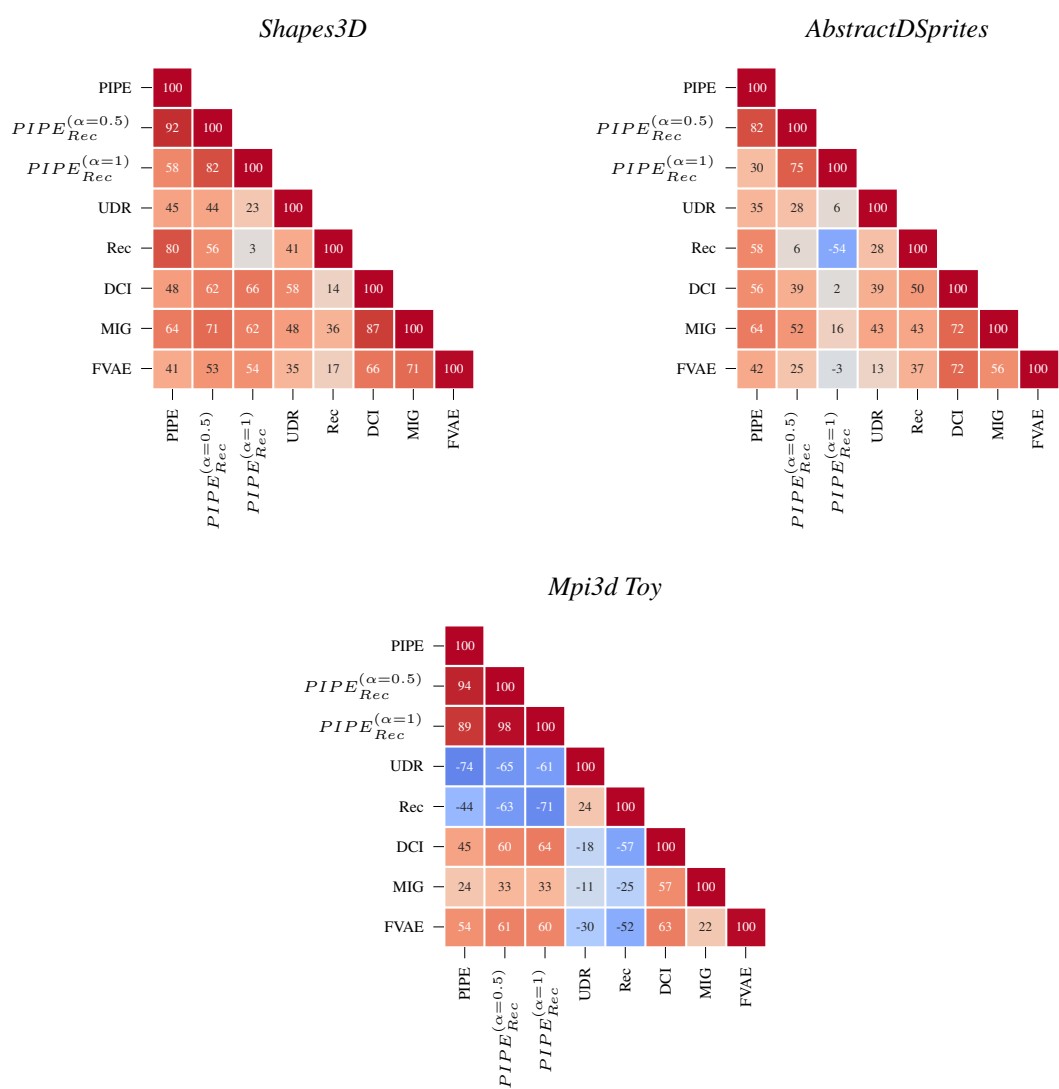

Figure 18: Spearman rank correlation between various metrics on three different datasets, similar as Figure 3. We additionally include $PIPE_{Rec}$ for $\alpha = 0.5$ and $\alpha = 1$.

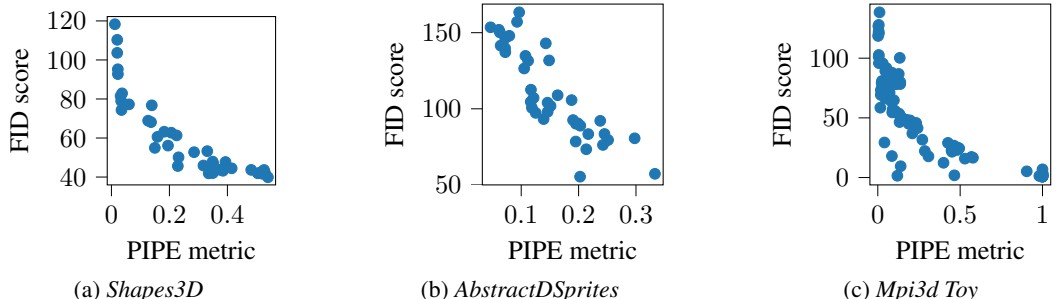

(a) *Shapes3D*    (b) *AbstractDSprites*    (c) *Mpi3d Toy*

Figure 19: FID vs PIPE metric quantifying the similarity between $D_{EP}$ and $D_{FP}$ on three different datasets over all evaluated models. If both metrics would correlate perfectly, one would expect a line from top left to bottom right corner.

VAE enough latent capacity to capture the complexity of the dataset. More specifically, we set z_dim to 20 for *Mpi3d Toy* and *Mpi3d Real*, and to 50 for *CelebA* respectively.

DAVA uses the same architecture and general hyperparameters as the baseline models. The discriminator of DAVA uses the same architecture as the encoder, with the last FC layer having a single output. To stabilize the adversarial training procedure of DAVA, we had to make use of a few measures more commonly used in GAN training. We used label smoothing as well as mixed batches for the training of the discriminator. We clipped the gradient norm for both the VAE and the discriminator at 1. As specified in Table 8, we had to use a lower weight for applying the adversarial gradient of the discriminator to the decoder. A bigger weight negatively affected reconstruction performance of the VAE. We applied instance normalization to the discriminator, which improved the adversarial gradient signal. We experimented with other types of normalization (batch, spectral, layer) for encoder, decoder and discriminator, but did not find a consistent improvement for any of them. Further, we improved the mechanism to control the KL-Divergence of a VAE proposed by (Burgess et al., 2017). Instead of using the absolute difference, we propose to take the difference to the power of 4. This allows the KL-Divergence to fluctuate in between batches, but strongly punishes big deviations.

We summarize the training procedure of DAVA in Algorithm 1. We use dimension-wise permutation ($PermuteDims$) to approximate the factorial posterior FP, similar to FactorVAE (Kim & Mnih, 2018) and as explained in B.2. The process of adjusting $C$ based on the accuracy of the discriminator is described in Algorithm 2.

| Encoder / DAVA Discriminator | Decoder |
|---|---|
| Input: [64,64,num channels] | FC, 256 ReLU |
| 4x4 conv, 2 strides, 32 ReLU | FC, 4x4x64 ReLU |
| 4x4 conv, 2 strides, 32 ReLU | 4x4 upconv, 2 strides, 64 ReLU |
| 4x4 conv, 2 strides, 64 ReLU | 4x4 upconv, 2 strides, 64 ReLU |
| 4x4 conv, 2 strides, 64 ReLU | 4x4 upconv, 2 strides, 64 ReLU |
| FC 256, FC $2 \cdot z\_dim$ | 4x4 upconv, 2 strides, num channels |

Table 4: VAE architecture used for all models in this study.

| Parameter | Value |
|---|---|
| Batch size | 128 |
| Optimizer | Adam |
| Adam: beta1 | 0.9 |
| Adam: beta2 | 0.999 |
| Adam: epsilon | 1e-8 |
| Learning rate | 0.0001 |
| Training steps | 150000 batches |
| Max. gradient norm | 1 |
| AnnealedVAE: $\gamma$ | 1000 |
| AnnealedVAE: iteration threshold | 100000 batches |
| z_dim | 10 |

Table 5: General hyperparameters.

| Parameter | Value | Discriminator |
|---|---|---|
| | | FC, 1000 leaky ReLU |
| Optimizer | Adam | FC, 1000 leaky ReLU |
| Adam: beta1 | 0.5 | FC, 1000 leaky ReLU |
| Adam: beta2 | 0.9 | FC, 1000 leaky ReLU |
| Adam: epsilon | 1e-8 | FC, 1000 leaky ReLU |
| Learning rate | 0.0001 | FC, 1000 leaky ReLU |
| | | FC, 2 |

Table 6: Hyperparameters and architecture of the discriminator of FactorVAE

| Model | Parameter | Value |
|---|---|---|
| $\beta$-TCVAE | $\beta$ | $[1, 2, 4, 8, 16, 32]$ |
| FactorVAE | $\gamma$ | $[5, 10, 20, 30, 50, 100]$ |
| AnnealedVAE | $C$ | $[\frac{1}{2}, 1, 2, 5, 25, 50]$ |
| ControlVAE | $C$ | $[16, 18, 25, 35, 180, 200]$ |

Table 7: Hyperparameter range explored for each model

| Parameter | Value |
|---|---|
| Discriminator: Optimizer | Adam |
| Discriminator: Adam: beta1 | 0.9 |
| Discriminator: Adam: beta2 | 0.999 |
| Discriminator: Adam: epsilon | 1e-8 |
| Discriminator: Learning rate | 0.0001 |
| Discriminator: Max. gradient norm | 1 |
| Dis encoder weight $\mu_{enc}$ | 0.3 |
| Dis decoder weight $\mu_{dec}$ | 0.001 |
| $\gamma$ | 500 |
| $\Delta C$ | 4e-5 |

Table 8: DAVA hyperparameters

## C.2 INDIVIDUAL CONTRIBUTION OF DIFFERENT BUILDING BLOCKS OF DAVA

One might wonder which building blocks of DAVA contribute most to the performance. Results during development show that the adversarial loss of DAVA alone delivers comparable performance to $\beta$-TCVAE and FactorVAE with the same issue of regularization strength being dependent on the dataset. Interestingly, for AnnealedVAE, the optimal $C$ for different datasets closely correspond to the values discovered by DAVA displayed in Figure 6. This also reflects the performance one could expect from DAVA without the adversarial loss.

## C.3 COMBINATION OF DAVA WITH EXISTING METHODS

The main contribution of DAVA lies in its ability to adapt the latent capacity bottleneck with respect to the current state of the model, evaluated by the discriminator. In that regard, it would be possible to use a different discriminator as the one proposed by DAVA. One such option is the discriminator used in FactorVAE, which could be used in a similar manner as the discriminator in DAVA to guide the latent capacity bottleneck. This would still allow for a dataset-specific annealing scheme without the need for dataset-specific hyperparameters.

---

**Algorithm 1** Training DAVA

---

$\theta_{enc}, \theta_{dec}, \theta_{dis} \leftarrow$ initialize parameters
**while** Not Converged **do**
    $x \leftarrow$ sample random mini-batch from $\tilde{D}$
    $\hat{z} \leftarrow Enc(x)$
    $\hat{x} \leftarrow Dec(\hat{z})$
    $\mathcal{L}_{vae} \leftarrow \log p(\hat{x}|\hat{z}) - \gamma(\mathrm{KL}(q(\hat{z})||p(\hat{z})) - C)^4$
    $\theta_{enc} \overset{+}{\leftarrow} -\nabla_{\theta_{enc}}\mathcal{L}_{vae}$
    $\theta_{dec} \overset{+}{\leftarrow} -\nabla_{\theta_{dec}}\mathcal{L}_{vae}$                                          $\triangleright$ Update VAE
    $\hat{z} \leftarrow Enc(x)$
    $\hat{x} \leftarrow Dec(\hat{z})$                             $\triangleright$ Recreate $\hat{z}$ and $\hat{x}$ after update
    $\tilde{x} \leftarrow Dec(PermuteDims(\hat{z}))$
    $acc \leftarrow accuracy(Dis, \hat{x}, \tilde{x})$
    $\theta_{dis} \overset{+}{\leftarrow} -\nabla_{\theta_{dis}}\log(Dis(\hat{x})) + \log(1 - Dis(\tilde{x}))$             $\triangleright$ Update Discriminator
    $C \overset{+}{\leftarrow} update_C(acc)$
    $\mu_{base} \leftarrow max((acc - 0.5) \cdot 100, 0)$         $\triangleright$ Weight grows linearly with higher accuracy
    $\theta_{enc} \overset{+}{\leftarrow} -\nabla_{\theta_{enc}}\mu_{base} \cdot \mu_{enc} \cdot \log(Dis(\hat{x}))$
    $\theta_{dec} \overset{+}{\leftarrow} -\nabla_{\theta_{dec}}\mu_{base} \cdot \mu_{dec} \cdot \log(1 - Dis(\hat{x}))$      $\triangleright$ Update VAE with adversarial loss
**end while**

---

**Algorithm 2** $update_C$

---

**Input:** accuracy $acc$
**if** $acc \leq 0.5$ **then**
    $U \leftarrow \Delta C$                           $\triangleright$ Only increase $C$ if discriminator is clueless
**else if** $acc \leq 0.51$ **then**
    $U \leftarrow 0$                                          $\triangleright$ Grace period
**else**
    $U \leftarrow -\Delta C$                        $\triangleright$ Decrease $C$ if discriminator gets better
**end if**
**Output:** update $U$

---

# D    COMPLETE QUANTITATIVE RESULTS OF ALL EVALUATED DATASETS

We present results for a large range of datasets. We also include results for the best regularization strength for each of the baseline approaches after supervised hyperparameter selection in the upper part of each table. This serves as an additional reference on what can be expected from the completely unsupervised approaches in a best case scenario. We further include the results of Recursive Disentanglement Network (RecurD) by Chen et al. (2021). These are to be interpreted with a grain of salt, as we were not able to reproduce them ourselves, but had to take the values from their paper. This means that first, only a limited set of datasets and metrics were evaluated. Second, there will be differences in model architecture, hyperparameter choice, the range of hyperparameter tuning, general training loop and possibly more things. Please note that there are two aspects to take into consideration in regard of the strong performance of AnnealedVAE. First, the selected range of hyperparameters is especially strong, as we include the hyperparameters discovered by DAVA. Second, as explained earlier for DAVA, we use $(\cdot)^4$ instead of the absolute value of the difference for controlling the KL-divergence. Further note that the exceptionally high PIPE metric score of FactorVAE on *NoisyDSprites* is caused by the decoder of the FactorVAE models always producing the same output regardless of the latent space. This is a design decision of PIPE, as described in Subsection B.3. Perhaps surprisingly, these models still seem to convey a small amount of information in the latent space as reflected by the supervised metrics. The same occurred with FactorVAE for *Mpi3d Toy*, interestingly though not on *Mpi3d Real*.

Table 9: Results *Shapes3D*

| Architecture | MIG | DCI | FVAE | PIPE | Rec |
|---|---|---|---|---|---|
| Best $\beta$-TCVAE ($\beta = 32$) | 0.39±0.09 | 0.65±0.03 | 0.76±0.06 | 0.36±0.03 | 0.0041±0.0008 |
| Best FactorVAE ($\gamma = 30$) | 0.39±0.16 | 0.54±0.12 | 0.79±0.05 | 0.30±0.08 | 0.0032±0.0005 |
| Best AnnealedVAE ($C = 1$) | 0.63±0.04 | 0.78±0.05 | 0.94±0.04 | 0.49±0.05 | 0.0027±0.0002 |
| Best ControlVAE ($C = 18$) | 0.08±0.03 | 0.20±0.06 | 0.68±0.12 | 0.02±0.00 | 0.0019±0.0004 |
| Best RecurD (Chen et al., 2021) | (0.31) | (0.58) | - | - | (0.0083) |
| Mean $\beta$-TCVAE | 0.26±0.06 | 0.50±0.03 | 0.76±0.03 | 0.11±0.03 | 0.0023±0.0004 |
| Mean FactorVAE | 0.20±0.10 | 0.41±0.07 | 0.78±0.04 | 0.22±0.04 | 0.0027±0.0005 |
| Mean AnnealedVAE | 0.52±0.02 | 0.70±0.02 | **0.92±0.01** | 0.26±0.03 | 0.0020±0.0001 |
| Mean ControlVAE | 0.07±0.03 | 0.17±0.04 | 0.68±0.07 | 0.02±0.00 | **0.0019±0.0004** |
| Ours | **0.62±0.05** | **0.78±0.03** | 0.82±0.03 | **0.61±0.04** | 0.0029±0.0004 |

Table 10: Results *AbstractDSprites*

| Architecture | MIG | DCI | FVAE | PIPE | Rec |
|---|---|---|---|---|---|
| Best $\beta$-TCVAE ($\beta = 4$) | 0.17±0.04 | 0.22±0.03 | 0.55±0.03 | 0.11±0.02 | 0.0017±0.0001 |
| Best FactorVAE ($\gamma = 50$) | 0.18±0.07 | 0.26±0.04 | 0.65±0.05 | 0.23±0.04 | 0.0019±0.0001 |
| Best AnnealedVAE ($C = 2$) | 0.23±0.04 | 0.26±0.04 | 0.65±0.05 | 0.13±0.02 | 0.0018±0.0001 |
| Best ControlVAE ($C = 18$) | 0.06±0.01 | 0.11±0.03 | 0.50±0.05 | 0.07±0.01 | 0.0016±0.0000 |
| Mean $\beta$-TCVAE | 0.12±0.01 | 0.18±0.01 | 0.45±0.02 | 0.19±0.01 | 0.0027±0.0000 |
| Mean FactorVAE | 0.12±0.02 | 0.19±0.02 | 0.55±0.03 | 0.20±0.03 | 0.0018±0.0000 |
| Mean AnnealedVAE | 0.15±0.01 | 0.21±0.01 | 0.54±0.02 | 0.18±0.01 | 0.0022±0.0000 |
| Mean ControlVAE | 0.04±0.02 | 0.07±0.01 | 0.42±0.02 | 0.06±0.01 | **0.0016±0.0000** |
| Ours | **0.23±0.04** | **0.27±0.05** | **0.67±0.05** | **0.35±0.03** | 0.0020±0.0002 |

Table 11: Results *Mpi3d Toy*

| Architecture | MIG | DCI | FVAE | PIPE | Rec |
|---|---|---|---|---|---|
| Best $\beta$-TCVAE ($\beta = 2$) | 0.23±0.04 | 0.33±0.01 | 0.42±0.06 | 0.07±0.01 | 0.0005±0.0000 |
| Best FactorVAE ($\gamma = 5$) | 0.10±0.05 | 0.20±0.02 | 0.46±0.03 | 0.36±0.16 | 0.0021±0.0003 |
| Best AnnealedVAE ($C = \frac{1}{2}$) | 0.21±0.07 | 0.31±0.02 | 0.41±0.05 | 0.12±0.01 | 0.0005±0.0000 |
| Best ControlVAE($C = 16$) | 0.04±0.01 | 0.18±0.02 | 0.44±0.04 | 0.02±0.01 | 0.0003±0.0000 |
| Mean $\beta$-TCVAE | 0.11±0.02 | 0.23±0.01 | 0.39±0.02 | 0.09±0.01 | 0.0006±0.0000 |
| Mean FactorVAE | 0.02±0.01 | 0.13±0.00 | 0.38±0.02 | 0.99±0.10 | 0.0041±0.0000 |
| Mean AnnealedVAE | 0.07±0.03 | 0.23±0.02 | **0.50±0.02** | 0.02±0.01 | **0.0003±0.0000** |
| Mean ControlVAE | 0.04±0.01 | 0.17±0.02 | 0.43±0.04 | 0.03±0.01 | **0.0003±0.0000** |
| Ours | **0.12±0.09** | **0.30±0.03** | 0.41±0.04 | **0.21±0.03** | 0.0006±0.0000 |

Table 12: Results *DSprites*

| Architecture | MIG | DCI | FVAE | PIPE | Rec |
|---|---|---|---|---|---|
| Best $\beta$-TCVAE ($\beta = 32$) | 0.34±0.02 | 0.50±0.03 | 0.60±0.00 | 0.76±0.14 | 0.0081±0.0000 |
| Best FactorVAE ($\gamma = 50$) | 0.10±0.04 | 0.22±0.02 | 0.73±0.04 | 0.41±0.05 | 0.0098±0.0007 |
| Best AnnealedVAE ($C = \frac{1}{2}$) | 0.36±0.01 | 0.51±0.01 | 0.76±0.03 | 0.42±0.03 | 0.0060±0.0006 |
| Best ControlVAE ($C = 18$) | 0.06±0.02 | 0.13±0.03 | 0.67±0.11 | 0.24±0.03 | 0.0008±0.0000 |
| Best RecurD (Chen et al., 2021) | (0.27) | (0.38) | - | - | (0.0047) |
| Mean $\beta$-TCVAE | **0.27±0.05** | **0.45±0.06** | **0.84±0.03** | 0.44±0.08 | 0.0024±0.0001 |
| Mean FactorVAE | 0.14±0.06 | 0.22±0.05 | 0.72±0.07 | 0.31±0.03 | 0.0052±0.0002 |
| Mean AnnealedVAE | 0.18±0.06 | 0.28±0.05 | 0.77±0.04 | 0.32±0.05 | 0.0013±0.0001 |
| Mean ControlVAE | 0.05±0.02 | 0.13±0.01 | 0.69±0.06 | 0.20±0.08 | **0.0008±0.0000** |
| Ours | 0.24±0.09 | 0.42±0.14 | 0.74±0.11 | **0.51±0.09** | 0.0025±0.0005 |

Table 13: Results *Mpi3d Real*

| Architecture | MIG | DCI | FVAE | PIPE | Rec |
|---|---|---|---|---|---|
| Best $\beta$-TCVAE ($\beta = 2$) | 0.16±0.08 | 0.35±0.06 | 0.56±0.05 | 0.03±0.01 | 0.0004±0.0000 |
| Best FactorVAE ($\gamma = 30$) | 0.02±0.01 | 0.18±0.04 | 0.43±0.03 | 0.42±0.10 | 0.0031±0.0003 |
| Best AnnealedVAE ($C = 1$) | 0.12±0.08 | 0.30±0.07 | 0.49±0.08 | 0.08±0.01 | 0.0004±0.0000 |
| Best ControlVAE ($C = 25$) | 0.06±0.02 | 0.22±0.01 | 0.54±0.05 | 0.01±0.00 | 0.0003±0.0000 |
| Mean $\beta$-TCVAE | 0.08±0.04 | 0.22±0.03 | 0.26±0.03 | **0.23±0.07** | 0.0015±0.0001 |
| Mean FactorVAE | 0.05±0.01 | 0.19±0.01 | 0.42±0.02 | 0.12±0.07 | 0.0022±0.0003 |
| Mean AnnealedVAE | 0.09±0.03 | **0.29±0.02** | **0.52±0.05** | 0.03±0.01 | **0.0003±0.0000** |
| Mean ControlVAE | 0.04±0.02 | 0.20±0.03 | 0.52±0.06 | 0.01±0.00 | **0.0003±0.0000** |
| Ours | **0.11±0.05** | 0.27±0.03 | 0.48±0.05 | 0.20±0.04 | 0.0006±0.0000 |

Table 14: Results *Smallnorb*

| Architecture | MIG | DCI | FVAE | PIPE | Rec |
|---|---|---|---|---|---|
| Best $\beta$-TCVAE ($\beta = 4$) | 0.16±0.03 | 0.34±0.01 | 0.60±0.02 | 0.18±0.03 | 0.0037±0.0000 |
| Best FactorVAE ($\gamma = 5$) | 0.12±0.06 | 0.25±0.03 | 0.64±0.02 | 0.24±0.03 | 0.0054±0.0009 |
| Best AnnealedVAE ($C = \frac{1}{2}$) | 0.20±0.01 | 0.28±0.01 | 0.58±0.01 | 0.22±0.02 | 0.0037±0.0000 |
| Best ControlVAE ($C = 180$) | 0.24±0.01 | 0.28±0.01 | 0.62±0.04 | 0.03±0.00 | 0.0016±0.0001 |
| Mean $\beta$-TCVAE | 0.18±0.02 | 0.29±0.01 | 0.57±0.02 | 0.11±0.01 | 0.0032±0.0001 |
| Mean FactorVAE | 0.03±0.03 | 0.13±0.01 | 0.53±0.04 | 0.17±0.09 | 0.0101±0.0039 |
| Mean AnnealedVAE | 0.21±0.01 | 0.28±0.00 | 0.60±0.01 | 0.05±0.00 | 0.0020±0.0001 |
| Mean ControlVAE | 0.23±0.02 | 0.28±0.01 | 0.62±0.03 | 0.03±0.00 | **0.0016±0.0001** |
| Ours | **0.25±0.01** | **0.30±0.01** | **0.62±0.01** | **0.25±0.02** | 0.0030±0.0001 |

Table 15: Results *NoisyDSprites*

| Architecture | MIG | DCI | FVAE | PIPE | Rec |
|---|---|---|---|---|---|
| Best $\beta$-TCVAE ($\beta = 4$) | 0.08±0.03 | 0.21±0.04 | 0.63±0.09 | 0.32±0.01 | 0.0804±0.0000 |
| Best FactorVAE ($\gamma = 100$) | 0.01±0.00 | 0.10±0.05 | 0.38±0.09 | 1.00±0.00 | 0.0892±0.0000 |
| Best AnnealedVAE ($C = 50$) | 0.06±0.02 | 0.13±0.03 | 0.65±0.08 | 0.48±0.29 | 0.0802±0.0000 |
| Best ControlVAE ($C = 16$) | 0.02±0.01 | 0.08±0.01 | 0.48±0.06 | 0.43±0.29 | 0.0801±0.0000 |
| Mean $\beta$-TCVAE | 0.05±0.03 | 0.11±0.05 | 0.44±0.08 | 0.31±0.05 | 0.0810±0.0000 |
| Mean FactorVAE | 0.01±0.00 | 0.06±0.02 | 0.33±0.03 | (1.00±0.00) | 0.0892±0.0000 |
| Mean AnnealedVAE | **0.09±0.05** | 0.18±0.06 | **0.58±0.09** | 0.27±0.03 | 0.0805±0.0000 |
| Mean ControlVAE | 0.02±0.01 | 0.08±0.01 | 0.49±0.01 | 0.35±0.14 | **0.0801±0.0000** |
| Ours | **0.09±0.05** | **0.22±0.05** | 0.55±0.07 | **0.32±0.04** | 0.0806±0.0001 |

Table 16: Results *Cars3D*

| Architecture | MIG | DCI | FVAE | PIPE | Rec |
|---|---|---|---|---|---|
| Best $\beta$-TCVAE ($\beta = 16$) | 0.14±0.02 | 0.33±0.08 | 0.90±0.02 | 0.07±0.01 | 0.0059±0.0000 |
| Best FactorVAE ($\gamma = 20$) | 0.10±0.01 | 0.16±0.04 | 0.88±0.03 | 0.10±0.02 | 0.0083±0.0005 |
| Best AnnealedVAE ($C = 1$) | 0.12±0.01 | 0.29±0.05 | 0.89±0.02 | 0.15±0.01 | 0.0059±0.0001 |
| Best ControlVAE ($C = 16$) | 0.05±0.02 | 0.10±0.02 | 0.84±0.05 | 0.03±0.00 | 0.0045±0.0000 |
| Best RecurD (Chen et al., 2021) | (0.17) | - | - | - | (0.0132) |
| Mean $\beta$-TCVAE | 0.11±0.03 | 0.24±0.01 | 0.91±0.02 | 0.06±0.01 | 0.0049±0.0001 |
| Mean FactorVAE | 0.06±0.02 | 0.12±0.03 | 0.74±0.17 | **0.28±0.14** | 0.0170±0.0023 |
| Mean AnnealedVAE | 0.06±0.02 | **0.25±0.04** | 0.86±0.01 | 0.08±0.01 | 0.0047±0.0001 |
| Mean ControlVAE | 0.03±0.01 | 0.11±0.01 | 0.76±0.09 | 0.03±0.00 | **0.0045±0.0000** |
| Ours | **0.15±0.01** | 0.23±0.04 | **0.94±0.01** | **0.28±0.02** | 0.0074±0.0001 |

## E    FURTHER RESULTS ON REAL-LIFE DATASETS

We further report results of DAVA against the full range of baseline models on the *CompCars* dataset. As described in the dataset description in A.10, *CompCars* is an extremely challenging dataset. Performance is most likely limited by the model architecture used, as can also be seen by the bad FID and PIPE metric scores displayed in Figure 20.

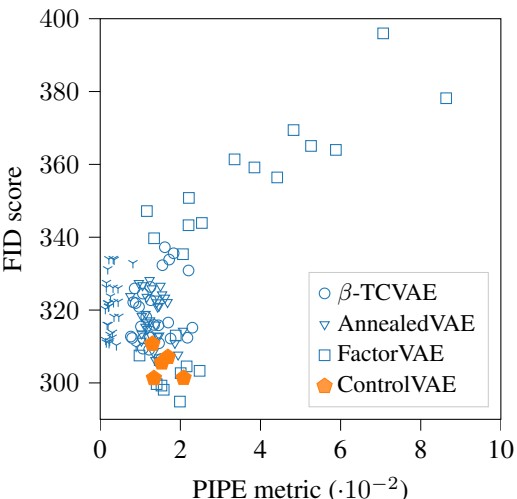

Figure 20: Similar to Figure 7, we display the FID of image samples from $\tilde{D}_{\text{FP}}$ of the respective models versus the original images of *CompCars*. DAVA achieves competitive FID scores even compared to models exploring a broad range of hyperparameters. FactorVAE models that achieve better disentanglement according to the PIPE metric suffer from significantly lower FID scores.

