# OpenReview forum: "DAVA: Disentangling Adversarial Variational Autoencoder"
_ICLR.cc/2023/Conference — ICLR 2023 poster_

### Official Review · Reviewer_SzgX · 2022-10-24

**Confidence:** 4
**Correctness:** 4
**Technical Novelty And Significance:** 3
**Empirical Novelty And Significance:** 3
**Recommendation:** 6

**Clarity, Quality, Novelty And Reproducibility:**


Clarity: The paper is overall well-written.

Quality: The paper is of good quality (except for the above points).

Reproducibility: the paper provides both the code and the detailed hyper-parameter settings.

Novelty: The proposed approach is new. However, the key idea shares some similarities to prior work (see above), and more in-depth discussion on the benefit/difference is needed.


**Strength And Weaknesses:**


Strength:

* The paper is well-written and easy to read.

* The proposed disentanglement metric can be useful for future work.

* The proposed disentanglement approach shows promising results on several datasets.

Weaknesses:

* Important prior work is missing in the discussion and evaluation.
    - The paper considers UDR as the baseline unsupervised metric for experimental comparison. ModelCentrality (https://arxiv.org/pdf/1906.06034.pdf) is another unsupervised metric and it was shown that ModelCentrality correlates better with the supervised metrics and is better at selecting disentangled generative models (both VAEs and GANs) than UDR. Given that ModelCentrality is proposed more than 2 years ago and performs better than UDR, it is important to discuss and compare with ModelCentrality in the paper.

* Some details are missing in the main text.
    - For the experiments in Section 4, which generative model is used in the experiment, and how many models are used?
    - The paper discusses the automatic process for tuning C and mu during training. Does it mean that we still need to tune gamma manually?

Besides the above points, I have the following questions:

* PIPE depends on training a classifier (neural network). Would it be sensitive to the hyper-parameter choices (e.g., architecture, number of training iterations)? It is better to discuss it in the paper.

* The key idea of DAVA is similar to FactorVAE. The difference is that FactorVAE regularizes the distance between the posterior latent distribution and the factored posterior latent distribution in the latent space using a total variation loss, while DAVA regularizes the distance in the image space (after passing the latents through the decoder) using an adversarial loss. Why is the approach in DAVA better than FactorVAE? I understand that in DAVA the hyper-parameters can be automatically tuned based on the accuracy of the introduced discriminator (section 5), but we can also do the same if FactorVAE's total variation loss is used in place of L_{dis} (i.e., tuning C and mu according to the loss value of total variation).



**Summary Of The Paper:**

The paper proposes a new unsupervised metric for evaluating the disentanglement of generative models (named PIPE), and a new approach for training disentangled variational autoencoder (named DAVA). PIPE evaluates the distinguishability of the distribution of reconstructed samples and the distribution of generated samples from factored posterior latent distribution. DAVA involves minimizing the distance of the above two distributions. Experiments show that the PIPE correlates well with supervised metrics, and DAVA achieves state-of-the-art disentanglement results on various datasets.

**Summary Of The Review:**


The paper is well-written, and the proposed metric and approach could be useful for the community. However, due to the above problems, I cannot give a positive score at this point. I recommend the authors address the questions in the rebuttal, and I will adjust the score accordingly.

---

> ### Author Response · Authors · 2022-11-16
> **Thank you for your review.**
>
> Dear Reviewer SzgX
>
> We thank you for your review and your constructive feedback.  We are glad to hear that in general you feel like our paper is well written and that the proposed work could be useful for the community. We now answer your questions.
>  1. On missing ModelCentrality.
>
>   We agree that not including ModelCentrality (MC) as a baseline is a crucial shortcoming and we thank you for pointing this out. For this reason, we have evaluated MC and now include it as a baseline in our results.
>
>  Even though MC outperforms UDR, PIPE still considerably outperforms MC. We believe that this is partly due to the following reasons:
> 	- To evaluate similarity of two models, MC samples from the prior of a model, but the EP distribution of the model might be far away from the prior. PIPE does not have this problem because it samples from the EP and FP distributions.
> 	- To evaluate similarity of two models, the second model is asked to encode generated samples of the first model (very similar to \tilde(D)_{EP}). But this distribution might be far away from the initial data distribution the encoder of the second model is used to.
> 	- If only a very small number of models actually find a disentangled representation, it is hard for MC to identify these models. PIPE can identify even a single disentangled model among many.
> 	- Due to implicit biases in model architecture, it is possible that models learn similar entangled representations, thus breaking the core assumption of MC that only disentangled representations are similar.
>
>  We have added a corresponding discussion section in Appendix B.5.
>
> 2. On details for experiments in Section 4.
>
>  We have included these details in Appendix B.1. We used disentanglement-lib (Locatello et al., 2019b), enabling easy reproducibility of our results, to train all models for Shapes3D, AbstractDSprites and Mpi3d Toy. We focus on FactorVAE and β-TCVAE as they are the models generally achieving the highest disentanglement performance in Locatello et al. (2019b). Following (Locatello et al., 2019b) on a reduced scale, we trained 5 random seeds each for 6 different hyperparameters. This leads to a total of 60 model instances per dataset, spanning a broad range of disentanglement performance.
>
> 3. On the manual tuning of gamma.
>
>  The influence of gamma is marginal compared to the other hyperparameters. We have used the same gamma for all evaluated datasets without issues and therefore deem it not necessary to tune gamma for new datasets.
>
> 4. On PIPEs sensitivity to hyper-parameters.
>
>  PIPE is moderately sensitive to hyper-parameter choices, mostly the number of training iterations. We suggest that ideally the architecture (and key hyperparameters like learning rate) of the model used in the PIPE metric should be similar to the ones used for the encoder in the VAE. The number of training iterations should be chosen to avoid overfitting of the discriminator.
>
>  An alternative option would be to quantify \omega using a metric such as FID. This would alleviate the hyperparameter problem but constrains PIPE to only be applied on domains where such pretrained models are available. We discuss this in more detail in Appendix B.4.
>
> 5. On the distinction between FactorVAE and DAVA.
>
>  You are correct, it would also be possible to adapt FactorVAE to enable the dynamic hyperparameter tuning. In that regard DAVA does not have an advantage. The main difference is that compared to FactorVAE, DAVA also allows the weights of the decoder to be optimized against the adversarial loss. FactorVAE mainly encourages low total correlation in the latent space, completely ignoring the decoder. According to experiments we did, this low total correlation correlates much worse with disentanglement than PIPE, evaluated in the image space. This leads us to believe that it is beneficial to have the adversarial loss in the image space.

---

> > ### Comment · Reviewer_SzgX · 2022-11-21
> > **Thank you!**
> >
> > Thank you very much for adding the experiments and answering the questions. The analysis/hypothesis you gave makes sense to me. One suggestion is to add these discussions (especially on PIPEs sensitivity and the distinction between FactorVAE and DAVA) to the paper.
> >
> > Based on these clarifications, I will increase the score from 5 to 6.

---

### Official Review · Reviewer_GT7q · 2022-10-24

**Confidence:** 4
**Correctness:** 3
**Technical Novelty And Significance:** 3
**Empirical Novelty And Significance:** 3
**Recommendation:** 6

**Clarity, Quality, Novelty And Reproducibility:**

* As explained above, the main text is really clear apart from the few mentioned issues.
* Quality is high, with a good set of datasets and clearly executed experiments.
* Even though some ideas were already present in the literature this is novel enough work in my opinion.


**Strength And Weaknesses:**

1. It might be good to mention that this really only allows for unidimensional disentangling?
   1. You mention this fact with DCI, but it might be the case that more complex datasets require correlated subspaces, which this metric/regularisation would discourage?
   2. What happened in your case for the DAVA model, say on AbstractDSprites? How was color or angle represented?
2. You do not comment on how to sample from D_FP vs D_EP, and on the potential difference in computational cost of both in the main text.
   1. When looking in the Appendix at Algorithm 1, one can see that you obtain \bar{z} simply by permuting the dimensions of z, but that wasn’t clear from reading the main text.
   2. It also doesn’t seem to be strictly the same as what the definition of D_FP implies, i.e. I was wondering if/when you would compute the marginal over the whole dataset?
   3. This should probably be expanded upon, as it is quite critical.
3. I found Figure 6 interesting, and the overall idea of automatically adapting the capacity of the VAE based on this schedule is a nice addendum.
   1. It might be worth comparing it to more standard techniques however, like GECO [1] or simpler heuristics like [2]
4. The abstract and start of the introduction is clear, and I found the related work to be quite thorough as well. However, I found the end of the introduction, which introduces the main idea of the paper, to be slightly hard to follow (despite being quite clear afterwards).
   1. More precisely, the section from “if a VAE learned a truly disentangled [...]” till “is a necessary condition for disentanglement” is too compressed to be understood on a first read. It now makes sense to me after having read the paper, but I would recommend simplifying/reworking it.
5. Section 3 is strong and well-derived, I felt like Figure 2 supported the arguments very clearly too. Similarly, Section 4 and 5 are strong and clear and contained all the experiments and clarifications I wanted to see.


[1] https://arxiv.org/abs/1810.00597
[2] https://arxiv.org/abs/2002.07514

**Summary Of The Paper:**

This paper proposes a novel way to encourage disentangling of representations for VAE-like models.

They introduce 2 distributions over samples x, D_EP which is the empirical distribution one would usually use with a VAE (where $q(z) = E[q(z | x)]$), and D_FP, which replaces the distribution over z with its forced factorial version $\bar{q}(z) = \prod_i q(z_i)$. One can generate samples and images from either distributions, and how much they differ is a signal for disentanglement.

This can then be used as a metric (which they call PIPE, Posterior Indifference Projection Equivalence), or as an extra loss, by using a Discriminator to penalize a model where these two distributions differ. They also show how one can use the same metric to control the capacity of a model on the fly, which is a good addendum as well. Finally, they show good results on a good set of well accepted simple benchmark datasets.

**Summary Of The Review:**

I found this work to be clear, well executed, it introduces several good ideas which I think the community can easily build upon, and presents good evidence of its capabilities, hence I would lean towards acceptance.

---

> ### Author Response · Authors · 2022-11-16
> **Thank you for your review.**
>
> Dear Reviewer GT7q
>
> We thank you for your review and your constructive feedback. We are glad to hear that you feel like our work is clear, well executed, and introducing several good ideas. We now answer your questions.
>
>  1. On unidimensional disentangling.
>
>  This is correct, for PIPE correlated subspaces are discouraged. PIPE is strongly based on the assumption of independent ground truth factors. DAVA would on average learn more compact representations than some of the other baselines, for example \beta-TCVAE. It would often learn the represent color or angle in a single dimension, for example also in 3DShapes.
>
>  2. On sampling from D_FP and D_EP.
>
>  We agree that this topic deserves a more in-depth explanation. For this reason, we have expanded Section B.2 in the appendix with relevant details. In summary, we follow the approach proposed in Algorithm 1 by (Kim & Mnih (2018)). This allows us to avoid computing the marginals over the complete dataset. We have put the corresponding reference to the appendix in the main text.
>
>  3. On Figure 6 and comparison to simpler baselines [1] and [2]
>
>  References [1] and [2] are very interesting on their own. They propose methods for dynamically tuning the weight \beta of the KL divergence. However, their focus lies in a better KL divergence / reconstruction loss trade-off, they do not put any focus on disentanglement. It is simpler to adapt the weight of the KL term according to the reconstruction loss, as the reconstruction loss can easily be computed during training. It would be interesting for future work to assess the effect these methods have on disentanglement.
>
>  4. On parts of the introduction being hard to follow
>
>  Thank you for the feedback, we have reworked that part and hope that it is easier to read now.
>
>  5. On Sections 3,4 and 5 being well-received.
>
>  We appreciate the positive feedback.

---

### Official Review · Reviewer_pitB · 2022-10-24

**Confidence:** 4
**Clarity, Quality, Novelty And Reproducibility:** The paper is well written but the nov…
**Correctness:** 4
**Technical Novelty And Significance:** 2
**Empirical Novelty And Significance:** 2
**Recommendation:** 6

**Strength And Weaknesses:**

* Strength
1. The introduced PIPE metric is interesting and somewhat novel.
2. Conduct extensive experiments on benchmark datasets to verify the effectiveness of the proposed approach

* Limitations
1. Technical innovation is somewhat limited. The proposed model is quite similar to the adversarial variational auto-encoders in prior work [Han et al. 2020] and [Carbajal et al. 2021] except for the metric PIPE. The defined EP and FP are not new concepts as they are already used in prior work such as FactorVAE and \beta-TCVAE.
2. Compared to more recent baselines. This work only compared the proposed approach with the baseline methods before 2020. It would be better to compare the proposed method with some latent baselines, such as ControlVAE [Shao et al 2020] and Recursive Disentanglement Network [Chen et al. 2021], and Jacobian regularization [Wei et al 2021]. For instance, ControlVAE also dynamically tunes the hyperparameters in the VAE objective function based on the output KL-divergence to improve the disentanglement.
3. Some latest works are missing. The authors only introduced the related work before 2019 while missing some latest work on disentangled representation learning, such as ControlVAE, Recursive Disentanglement Network, and Jacobian regularization as mentioned above.
4. There is a trade-off between reconstruction error and disentanglement. It would be better to compare the reconstruction error with that of prior work, such as FactorVAE and ControlVAE.

References:
[Chen et al. 2021] Recursive Disentanglement Network. In International Conference on Learning Representations, 2021.
[Wei et al 2021] Orthogonal jacobian regularization for unsupervised disentanglement in image generation. In Proceedings of the IEEE/CVF International Conference on Computer Vision (pp. 6721-6730).
[Han et al. 2020] Disentangled adversarial autoencoder for subject-invariant physiological feature extraction. IEEE signal processing letters, 27, 1565-1569.
[Carbajal et al. 2021] Disentanglement Learning for Variational Autoencoders Applied to Audio-Visual Speech Enhancement, https://arxiv.org/abs/2105.08970

**Summary Of The Paper:**

This work developed an adversarial variational auto-encoders to learn the disentangled representations of observed data in an unsupervised manner. Specifically, it introduced a new metric that computes the difference of the reconstruction between EP and FP to measure disentanglement. Experimental results on multiple datasets illustrated the effectiveness of the proposed method.

**Summary Of The Review:**

The technical contribution is somewhat limited since the proposed adversarial variational auto-encoders has been developed by the prior work. In addition, this work needs to compare to some latest baselines. The experimental results need to present the reconstruction error.

---

> ### Author Response · Authors · 2022-11-16
> **Thank you for your review.**
>
> Dear Reviewer pitB
>
> We thank you for your review and your constructive feedback. We are glad to hear that you find the PIPE metric interesting and are happy with our extensive experiments on the benchmark datasets. We now answer your questions.
>
>  1. On limited technical innovation and similarity to previous work.
>
>  While there is some similarity between [Han et al. 2020], [Carbajal et al. 2021] and our work, there are distinct and major differences. The largest difference is that both [Han et al. 2020] and [Carbajal et al. 2021] work in a semi-supervised setup, where they use an adversarial network to keep the mutual information between part of the latent space and a label minimal. This is vastly different from our approach, where we use an adversarial network in a completely unsupervised fashion to encourage disentanglement of the complete latent space and control over the latent space capacity bottleneck. Further, both [Han et al. 2020] and [Carbajal et al. 2021] apply an adversarial network to the latent space of a VAE, while our work applies the adversarial network to the output of the decoder, allowing then both encoder and decoder to be optimized against the adversary.
>
>  2. On comparisons to more recent baselines.
>
>  We appreciate the pointer to some interesting relevant work, which we will also include in the related work section. We included the ControlVAE loss into our experimental setup and run all baselines with a choice of main hyperparameters displayed as best-performing in their paper while keeping everything else consistent with the other baselines and DAVA. We want to mention that ControlVAE only tunes \beta to make the KL-divergence follow a predetermined path, similar to AnnealedVAE but with a more complex approach. This still includes the predetermined path as a major hyperparameter, and, as our results show, this hyperparameter needs to be tuned. It is very dataset-specific, as we have included all best-performing hyperparameters from the ControlVAE paper in our experiments and the disentanglement performance of ControlVAE is still the worst out of all baselines. Further exploration would be necessary to improve disentanglement performance for ControlVAE, making our point of the advantage of the automatic hyperparameter tuning of DAVA.
>
>  We were not able to reproduce Recursive Disentanglement Network (RecurD) as there was no code available. We include the limited results on disentanglement from the paper and compare it to DAVA. Keep in mind that RecurD results are achieved after supervised hyperparameter optimization and on a different architecture and training schedule than DAVA.
>
>  | Dataset  | Model  | MIG   	| DCI   	| Rec       	|
> |----------|--------|-----------|-----------|---------------|
> | Shapes3D | RecurD | 0.31  	| 0.58  	| 0.0083    	|
> |      	| DAVA   | 0.62±0.05 | 0.78±0.03 | 0.0029±0.0004 |
> | DSprites | RecurD | 0.27  	| 0.38  	| 0.0047    	|
> |      	| DAVA   | 0.24±0.09 | 0.42±0.14 | 0.0025±0.0005 |
> | Cars3D   | RecurD | 0.17  	| -     	| 0.0132    	|
> |      	| DAVA   | 0.15±0.01 | 0.23±0.04 | 0.0074±0.0001 |
>
>  Regarding the work by [Wei et al 2021] (OroJaR) on GANs, unfortunately according to the authors, OroJaR for VAEs is future work. We therefore are not able to include OroJaR as a baseline. The paper also does not include any results on commonly used disentanglement metrics which renders a direct comparison impossible.
>
>  3. On recent related work missing.
>
>  Thank you for pointing this out. We have added the relevant papers to our related work section.
>
>  4. On the trade-off between reconstruction error and disentanglement.
>
>  We agree that there usually is a trade-off between reconstruction error and disentanglement. We now provide reconstruction error for all experiments in the appendix. While optimizing that trade-off was not goal of our work, we can see that DAVA seems to achieve a similar trade-off than the baselines, if not even better. ControlVAE always achieves the lowest reconstruction error but is severely lacking in disentanglement performance.

---

> > ### Comment · Reviewer_pitB · 2022-11-21
> > **Thank you for your rebuttal**
> >
> > Thank you for adding recent works in the related works and doing additional experiments to compare your method with latest baselines. I will increase the score from 5 to 6.

---

### Author Response · Authors · 2022-11-16
**Paper updated**

Dear Reviewers

We are glad to see that you value PIPE as an "interesting metric" that could be "useful for future work". Regarding the training procedure for VAEs (DAVA), you value the "extensive experiments" that provide "good evidence of its capabilities".
Thanks to your feedback, we were able to improve the flaws of our work. In particular, we have changed the following:
 - We include ModelCentrality, another unsupervised disentanglement metric, as a baseline for comparison with PIPE.
 - We include ControlVAE as a baseline in all our experiments for comparison with DAVA.
 - We include results for Recursive Disentanglement Network (RecurD) (where possible) in the complete quantitative results in the Appendix.
 - We include reconstruction loss for the complete quantitative results in the Appendix.
 - We have rewritten part of the introduction for improved clarity.
 - We have extended our related work section with relevant work brought up in the reviews.
 - Minor changes in the manuscript.

We thank you for your reviews and hope that we were able to address the shortcomings of our previous version. We look forward to further discussions.

---

### Decision · Program_Chairs · 2023-01-20

**Decision:**

Accept: poster

**Justification For Why Not Higher Score:**

Novelty is not high (combinatorial ideas especially for the DAVA network architecture).

**Justification For Why Not Lower Score:**

Although novelty is not that high, in my opinion good experimental progress should not be easily dismissed.

**Metareview: Summary, Strengths And Weaknesses:**

This paper consider the task of disentangled presentation learning for generative models. The contributions are:
1. The PIPE metric as a new metric to evaluate the disentanglement of learned representations.
2. The DAVA architecture to achieve better disentanglement.
The proposed ideas are evaluated with multiple experiments with overall supportive results.

Reviewers find the PIPE metric and the DAVA architecture to be somewhat novel (i.e., either based on existing ideas or the novelty is on the combinatorial side). After revisions reviewers overall are happy with the newly added baselines and the extensive empirical evaluations.

**Note From Pc:**

if the above contains the word "oral" or "spotlight" please see: "oral" presentation means -> notable-top-5% and "spotlight" means -> notable-top-25%. As stated in our emails, we are disassociating presentation type from AC recommendations

**Summary Of Ac-Reviewer Meeting:**

N/A